# Gasification of Liquid Hydrocarbon Waste by the Ultra-Superheated Mixture of Steam and Carbon Dioxide: A Thermodynamic Study

Sergey M. Frolov [1,2,*], Konstantin S. Panin [2] and Viktor A. Smetanyuk [1]

1   Department of Combustion and Explosion, Semenov Federal Research Center for Chemical Physics, Moscow 119991, Russia; smetanuk@mail.ru
2   Institute for Laser and Plasma Technologies, National Research Nuclear University MEPhI, Moscow 115409, Russia; kostyapanin3@gmail.com
*   Correspondence: smfrol@chph.ras.ru

**Abstract:** The thermodynamic modeling of waste oil (WO) gasification by a high-temperature gasification agent (GA) composed of an ultra-superheated $H_2O/CO_2$ mixture is carried out. The GA is assumed to be obtained by the gaseous detonation of fuel–oxidizer–diluent mixture in a pulsed detonation gun (PDG). N-hexadecane is used as a WO surrogate. Methane or the produced syngas (generally a mixture of $H_2$, CO, $CH_4$, $CO_2$, etc.) is used as fuel for the PDG. Oxygen, air, or oxygen-enriched air are used as oxidizers for the PDG. Low-temperature steam is used as a diluent gas. The gasification process is assumed to proceed in a flow-through gasifier at atmospheric pressure. It is shown that the use of the detonation products of the stoichiometric methane–oxygen and methane–air mixtures theoretically leads to the complete conversion of WO into a syngas consisting exclusively of $H_2$ and CO, or into energy gas with high contents of $CH_4$ and $C_2$-$C_3$ hydrocarbons and an LHV of 36.7 (fuel–oxygen mixture) and 13.6 MJ/kg (fuel–air mixture). The use of the detonation products of the stoichiometric mixture of the produced syngas with oxygen or with oxygen-enriched air also allows theoretically achieving the complete conversion of WO into syngas consisting exclusively of $H_2$ and CO. About 33% of the produced syngas mixed with oxygen can be theoretically used for PDG self-feeding, thus making the gasification technology very attractive and cost-effective. To self-feed the PDG with the mixture of the produced syngas with air, it is necessary to increase the backpressure in the gasifier and/or enrich the air with oxygen. The addition of low-temperature steam to the fuel–oxygen mixture in the PDG allows controlling the $H_2$/CO ratio in the produced syngas from 1.3 to 3.4.

**Keywords:** gasification; organic waste; waste oil; ultra-superheated $H_2O/CO_2$ gasification agent; pulsed detonation gun; syngas; energy gas; self-feeding

## 1. Introduction

A variety of carbon-containing waste materials with different properties consisting predominantly of C, H, and O elements is commonly referred to as organic waste. The gasification of liquid/solid organic wastes with steam and carbon dioxide is considered a competitive and cost-effective waste processing technology [1,2], especially when the heat required for the processing is obtained by environmentally clean technologies (plasma [3], microwave [4], solar [5], etc.), other than waste incineration. The objective of gasification is to completely convert the carbon contained in the waste. The gasification process generally includes waste drying, pyrolysis, and thermal cracking followed by the partial oxidation of the produced gases, tar, and char at higher temperatures, leading to the formation of a product gas (a mixture of $H_2$, CO, $CH_4$, $CO_2$, etc.) commonly referred to as syngas or energy gas depending on its further application. The process includes multiple heterogeneous/homogeneous exothermic/endothermic reactions between molecules, atoms,

and active radicals as well as ionized and electronically excited species in case of plasma gasification. The efficiency of the gasification process is high, with a high yield of the produced gas and the less carbon remains in the by-products (a high completeness of carbon conversion into CO). Thermodynamic calculations indicate [6] that at gasification temperatures ~600 °C, carbon and oxygen are present as $CO_2$, tar, and char; at temperatures above ~900 °C, $CO_2$ breaks down to CO in the presence of carbon and the available oxygen mostly reacts with the carbon to form CO and $CO_2$; and at temperatures above ~1500 °C tar and char are completely transformed to syngas or energy gas composed mainly of $H_2$ and CO or $CH_4$ and CO, respectively. As for the effect of gasification pressure, the increase in pressure at a fixed gasification temperature of 1000 °C decreases the mole fractions of $H_2$ and CO in the produced gas and increases those of $CO_2$ and $CH_4$ [7]. A similar trend exists at temperatures above 1500 °C but the differences in product yields look negligible.

The use of steam and/or $CO_2$ as a gasifying agent (GA) has a number of advantages [8,9]. Firstly, the produced gas is not diluted with other gases. Secondly, waste gasification with steam/$CO_2$ requires less GA due to their high enthalpies. Thirdly, the use of an $H_2O/CO_2$ mixture allows controlling the composition of the produced gas. Fourthly, the use of steam as GA increases economic efficiency [10,11]. Fifthly, in the absence of free oxygen, the produced gas does not contain dioxins and furans, which facilitates gas purification operations [12]. Sixth, the amount of $H_2$ produced by steam gasification of waste is several times greater than by air gasification [13].

Depending on the level of the gasification temperature, all gasification technologies are divided into low-temperature and high-temperature technologies [14]. Low-temperature gasification is usually carried out at temperatures below 1000 °C and produces gas, char, and slag. In the literature, there are several noteworthy thermodynamic studies on the low-temperature steam gasification of organic wastes. Thus, the thermodynamic calculations of the steam gasification of woody biomass and cellulose at atmospheric pressure using the minimization of Gibbs free energy were reported in [15,16]. Based on the results of the models, the optimal ranges of the steam–feedstock (S/F) ratio were recommended. Various modeling approaches to simulate the steam gasification of biomass with regard to chemical and physical kinetic limitations were investigated in [17]. An equilibrium gas–solid model based on the minimization of the Gibbs free energy was developed in [18] for estimating the theoretical yield and the equilibrium composition of the syngas produced from a biomass during various thermochemical conversion processes (pyrolysis, partial oxidation, and gasification). The results of the calculations of the thermodynamic equilibrium state for a system initially composed of biomass and water for evaluating the influence of the gasification temperature, pressure, S/F ratio, and the type of biomass on the efficiency of the gasification system in terms of several criteria related to syngas yield and quality were reported in [19]. The equilibrium model of steam gasification for predicting the performance of $H_2$-rich gas production from biomass was reported in [20] and used to compare model predictions with experimental data. The arising discrepancies in product gas composition were explained by the lack of equilibrium conditions in a gasifier. The model was modified by correcting the equilibrium constants of several reactions by multiplying each by a pre-factor. The Aspen Plus software was applied in [21–23] to investigate the steam gasification of biomass in terms of perspectives in $H_2$ production and in [24] to study the atmospheric pressure steam co-gasification of wet sewage sludge waste and torrefied biomass using the nonstoichiometric thermodynamic equilibrium model with the Gibbs free energy minimization. For co-gasification, the water in the sewage sludge waste acted as the gasification agent. The optimal condition and blending ratio were determined by the maximum $H_2$ yield.

The main disadvantages of all the existing technologies of low-temperature gasification with $H_2O/CO_2$ are the low quality of the produced gas due to the high content of tar and $CO_2$, the low efficiency of gasification due to the large amount of remaining char, the difficulty of managing the quality of the produced gas due to the long residence time of feedstock in a gasifier, as well as the low yield of the produced gas due to its partial use

in the production of heat for gasification. Modern R&D in the field of low-temperature gasification is mainly aimed at feedstock pre-processing and increasing its reactivity by adding catalysts [25].

High-temperature gasification is usually carried out at temperatures above 1200 °C, which are achieved using combustion processes, as well as plasma and solar radiation [26–28]. In the literature, there are several noteworthy thermodynamic studies on the high-temperature steam gasification of organic wastes. The results of the thermodynamic calculations for the high-temperature gasification of organic feedstock aimed at determining the maximum conversion efficiency when all carbon was oxidized to CO were reported in [29,30]. Calculations were made for wood and pyrolytic oil with added $CO_2$ and/or $H_2O$. Both the wood and oil produced syngas with the joint content of $H_2$ and CO close to 100% at temperatures above 930 °C. The contents of $H_2$ and CO at the steam gasification of the oil were 62 and 38 vol%., i.e., the $H_2$/CO ratio was about 1.6. The results of the thermodynamic calculations on the plasma-assisted gasification of a biomass (wood) considering air, $CO_2$, and $H_2O$ as plasma-forming gases were presented in [31]. The calculated values of the $H_2$/CO ratio varied from 0.64 to 1.07 for air plasma, from 0.18 to 1.07 for $CO_2$ plasma, and from 1.07 to 3.65 for $H_2O$ plasma. The results of the thermodynamic calculations of the high-temperature steam gasification of municipal solid waste were reported in [32]. The calculations were made for temperatures up to 2700 °C at atmospheric pressure without accounting for energy loss. The yield of syngas increased with the temperature attaining a nearly constant value above 930 °C. Solid-phase carbon was completely transformed to CO in the gas phase in these conditions. The maximum yield of syngas reached 94.5 vol% (60.9% $H_2$, 33.6% CO). The content of oxidants at high temperatures was very low. The content of HCl varied from 1.2 to 1.6 vol%. Sulfur was represented by $H_2S$ up to 1630 °C, but dissociated into S and H atoms with increasing temperature. At temperatures above 1330 °C, $CaCl_2$, Fe, SiO, and Cl with a total content of less than 1 vol% appeared in the gas phase. This ensured 100% carbon conversion. The mineral part of the feedstock in the temperature range 930–1930 °C was mainly represented by $SiO_2$, $CaSiO_3$, $Fe_3C$, and Fe, but completely passed into the gas phase at temperatures above 1930 °C, forming the corresponding gaseous compounds. Importantly, there were no harmful impurities in the gas and condensed products of the high-temperature steam gasification of municipal solid waste. The low heating value (LHV) of the syngas obtained by steam gasification was 19.4 MJ/kg. The thermodynamic calculations of the high-temperature steam gasification of various organic feedstocks were performed in [33]. The amount of steam added was equal to that required for the stoichiometric gasification of 1 kg of feedstock. The yield of syngas for all tested feedstocks at 1230 °C considered was 98–100%. For the plastics, the $H_2$/CO ratio was equal to 2, and the syngas LHV was 11.6 MJ/nm$^3$. The use of $H_2O$ as a gasifying agent for textile provided syngas with an LHV of 11.3 MJ/nm$^3$. For the wood sawdust, the syngas LHV was 11.3 MJ/nm$^3$. The authors claimed that the calculated gas composition and the LHV of municipal solid waste and wood sawdust corresponded well to the experimental data obtained in plasma reactors and the arising differences (10 to 15%) were attributed to energy losses not included in the thermodynamic calculations.

The main advantages of the high-temperature $H_2O$/$CO_2$ gasification technologies are the produced high-quality gas due to the absence or very low content of tar and $CO_2$, high gasification efficiency due to the absence or small residues of tar and char, the ease of quality control of the produced gas due to the short residence time of feedstock in a gasifier, and the high yields of the produced gas due to the use of external energy sources for delivering the heat required for gasification. In addition to these advantages, existing high-temperature plasma and solar-assisted technologies have certain limitations. Despite the fact that the typical operation temperature of plasma gasifiers is below 2000 °C, plasma technologies require large amounts of electricity for gas–plasma transition. Moreover, plasma gasifiers need special materials with a refractory lining and water cooling, as well as short-service life arc electrodes. The main limitation of solar-assisted gasifiers is their intermittent nature.

Of particular interest is the new technology for the gasification of organic wastes by the ultra-superheated $H_2O/CO_2$ mixture with a temperature above 1500 °C [34]. The ability of such a GA to gasify liquid/solid organic wastes without a negative environmental impact is well known. Such a GA can be obtained by the gaseous detonation of fuel–oxidizer–diluent mixture in a pulsed detonation gun (PDG) operating in a pulse mode and periodically producing the strong shock waves and dense supersonic jets of high-temperature detonation products. At temperatures above 1500 °C, the tar and char formed at the initial stages of the gasification process are completely converted into syngas or energy gas, ideally consisting only of $H_2$ and CO or $CH_4$ and CO, respectively, in a proportion depending on the feedstock, while the cooled mineral residue consists of safe simple oxides and the aqueous solutions of oxygen-free acids, such as HCl, HF, $H_2S$, etc. and ammonia. The mineral residues can be used as additives to building materials, and the acids can be separated and concentrated. In other words, the new gasification technology potentially allows the complete processing of organic wastes into useful products without emissions into the atmosphere and water bodies.

The objectives of this paper are (i) to conduct the thermodynamic modeling of liquid waste oil (WO) gasification by a high-temperature GA obtained by the pulsed detonation of the stoichiometric methane–oxygen mixture; (ii) to determine the conditions for PDG self-feeding with the produced syngas; and (iii) to find out whether the gasification technology under consideration can be implemented with the replacement of oxygen with air or oxygen-enriched air, which would make the technology much more attractive and cost-effective. It is worth emphasizing that thermodynamic modeling generally provides the trends rather than the actual values of temperature and product composition. The differences between calculations and experiments are usually attributed to the imperfect mixing of components, finite rates of heat and mass transfer and chemical transformations, as well as thermal losses.

## 2. Materials and Methods

### 2.1. Technology

The essence of the new gasification technology is clear from considering Figure 1. Organic waste is continuously supplied to the flow-through gasifier where it is subjected to the pulse-periodic action of the ultra-superheated GA generated in the PDG, whereas gasification products continuously outflow from the gasifier and are transmitted to the customer either completely or partly (if a part of the gasification products is utilized for PDG self-feeding). The PDG is a tube with one end closed and the other open. The closed end of the tube is equipped with ports for supplying fuel, oxidizer, and low-temperature steam (110–120 °C) through the corresponding manifolds with control valves. The operation cycle begins with the supply of the combustible mixture components to the PDG. After the ignition of the combustible mixture with a spark plug followed by flame acceleration and deflagration-to-detonation transition, a detonation wave propagates in the mixture at a very high velocity (1700–2400 m/s), which transforms the mixture into detonation products, consisting mainly of steam and carbon dioxide at high temperatures and pressures. When the detonation wave exits through the open end of the tube into the gasifier, high-temperature detonation products expand into the gasifier in the form of a dense supersonic jet with a velocity on average above 1000 m/s, and the pressure in the PDG is reduced. When the pressure in the PDG drops to the pressure in the gasifier, a new portion of the components of the combustible mixture is supplied through the ports at the closed end of the tube. Once the PDG is filled with the combustible mixture, the spark plug fires and the next operation cycle starts. The detonation products of a hydrocarbon fuel–oxygen mixture can contain enough ultra-superheated $H_2O$ and $CO_2$ for the complete gasification of organic feedstock, therefore diluting the combustible mixture with low-temperature steam becomes unnecessary.

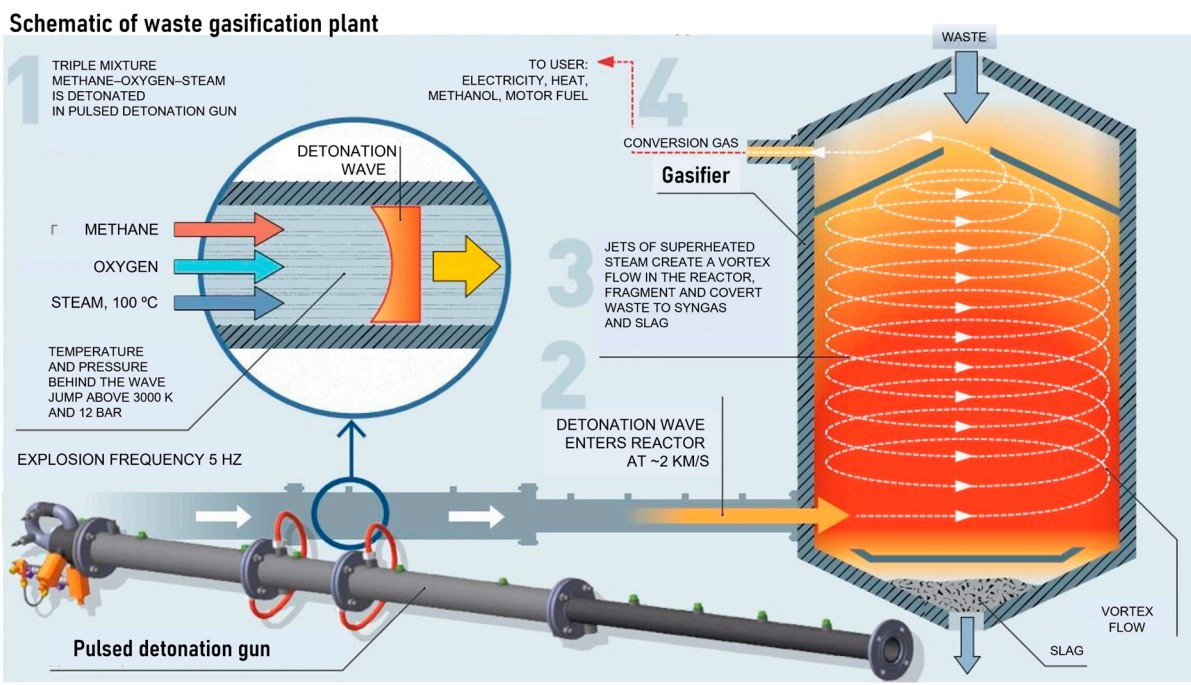

**Figure 1.** Schematic of waste gasification plant [14].

The GA generator (i.e., PDG) operates in a pulse mode, and the pulse frequency is mainly determined by the time of tube filling. The average parameters of the emanating GA jet (temperature, composition, velocity, etc.) are determined by the composition of the combustible mixture and the degree of its dilution with low-temperature steam. The PDG is connected with an open end to the gasifier of a compact geometric shape, which avoids the accumulation and slagging of waste. It is advisable to attach PDGs to the gasifier in pairs and place them coaxially opposite to each other to create strong counter shock waves and powerful vortex structures that increase the residence time of the waste particles and their carbonized residues. The pulsed shock waves emanating from the PDGs and having enormous destructive power fragment the waste into small particles and prevent their agglomeration and adhesion to gasifier walls. To increase the average residence time of the waste particles, a cascade of gasifiers can be used, which communicate with each other through tubes to allow gases and particles to flow from one gasifier to another. In all cases, the average operation pressure in the gasifier(s) should be above atmospheric pressure to prevent the suction of atmospheric air. In general, the operation process of the PDG–gasifier assembly has much in common with reciprocating engines. The gasifier can be made of conventional construction materials and be water-cooled as all physical and chemical processes mainly proceed far from the walls. The PDG is effectively cooled down from the interior during the filling process, and can also be water-cooled from the exterior with the further use of the heated water for producing the low-temperature steam required for the operation process. Currently, such installations for the steam gasification of organic waste, meeting specific requirements in terms of the operation temperature, the feedstock residence time, the composition of the produced syngas, etc., can be designed using computational fluid dynamics [35] and equilibrium gasification models [32,33,36]. The technology has been implemented in experimental installations and tested on a number of organic wastes, including WO [37], which is a subject of particular interest for environmentally safe utilization [38,39].

*2.2. Models*

Figure 2 shows two models of the gasification process. Both models include the supply of combustible mixture components to the PDG, the generation of GA in the detonation wave, the expansion of GA to the gasifier, the supply of the feedstock to the gasifier, and

the outflow of gasification products from the gasifier first to a cooler and then to a customer. Despite the similarity, there is an important difference between the models. In model 1 (Figure 2a), the PDG is fed by some selected fuel, oxidizer, and steam. In model 2 (Figure 2b), after the gasification process is established, the self-feeding of the PDG with the produced syngas is turned on, whereas the supply of the selected starting fuel is terminated. The gasifier outlet can be equipped with the pressure relief valve as shown in Figure 2a,b. After each detonation pulse in the PDG, the valve relieves the pressure in the gasifier when it exceeds a preset value. In the presence of an activated valve, the CJ detonation products expand from the PDG to the gasifier with the backpressure growing with time. Obviously, the growth rate of the backpressure depends on the PDG–gasifier volume ratio.

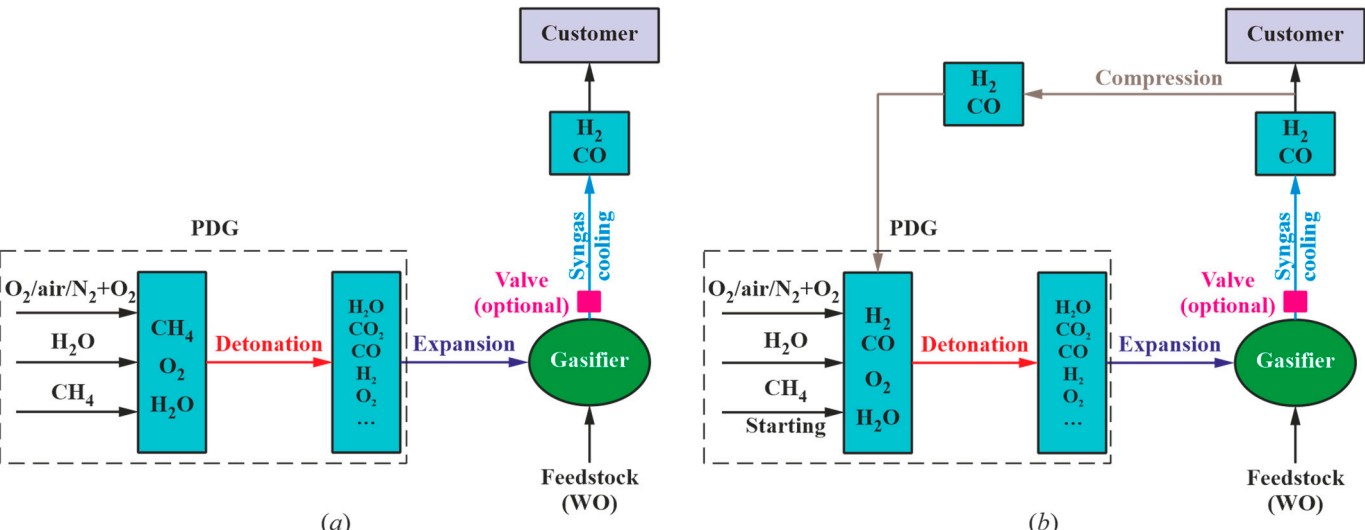

**Figure 2.** Two models of the WO gasification process: (**a**) Model 1 with the supply of WO to the gasifier without PDG self-feeding; (**b**) model 2 with the supply of WO to the gasifier with the self-feeding of the PDG with the produced syngas.

To fully model the gasification process, three-dimensional gas-dynamic calculations with the kinetic mechanisms of feedstock pyrolysis and oxidation are required. An example of such calculations is given in [35]. In this work, the thermodynamic modeling of the gasification process is carried out. For this purpose, the following simplifying assumptions are adopted:

(1) The fuel for PDG is methane (model 1) or methane as a starting fuel and the produced syngas as the main fuel (model 2). Oxygen, air, or oxygen-enriched air is used as an oxidizer for PDG. The mixture of fuel and oxidizer is stoichiometric, i.e., the fuel–oxidizer equivalence ratio $\Phi$ is $\Phi = 1$. The fuel–oxidizer mixture enters the PDG under normal conditions ($P_0 = 1$ bar, $T_0 = 300$ K). Steam with parameters $P_{0s} = 1$ bar and $T_{0s} = 400$ K can also be supplied to the PDG as a diluent.

(2) In models 1 and 2, feedstock gasification occurs at GA parameters (temperature, pressure, and composition), which are constant in time and correspond to the state of the Chapman–Jouguet (CJ) detonation products expanded to pressure $P$, which is normally equal to $P_0 = 1$ bar but can also attain an intermediate value $P_0 \leq P \leq P_{CJ}$ ($P_{CJ}$ is the CJ pressure) if the pressure relief valve is installed at the gasifier outlet. It should be borne in mind that this assumption underestimates GA parameters because the expansion of detonation products is a finite-time process.

(3) The WO is modeled by n-hexadecane ($C_{16}H_{34}$), which is often used as a physical and chemical surrogate for lubricant oils [40]. It is assumed that the WO enters the gasifier as a liquid at a temperature of $T_0 = 300$ K.

(4)  The mixing of the WO with GA occurs instantaneously. Since the final temperatures of the WO–GA mixture are significantly higher than the WO boiling point, only gas-phase mixing is considered.

(5)  The gasification of WO in GA occurs at a constant pressure *P* in the absence of heat and mass exchange with the ambience.

(6)  All reactions occur in the gas phase.

The simulation is carried out using the SDToolbox software [41] (to determine the CJ parameters of the detonation products) and the Cantera software [42] (to determine the thermodynamic parameters and composition of the produced syngas) and is divided into three stages. In the first stage, the GA parameters (the pressure, temperature, and composition of the CJ detonation products) are determined using the SDToolbox software. The equilibrium parameters of the detonation products expanded to pressure $P_0 \leq P \leq P_{CJ}$ are determined by an additional solution to problem $S, P = const$. In the second stage, the temperature and composition of the WO–GA mixture are calculated. The temperature of the mixture is determined by the temperatures of the components and the WO/GA mass ratio. In the third stage, the conversion of the resulting WO–GA mixture into syngas is considered in an adiabatic flow reactor at constant pressure $P$ ($H, P = const$ problem) and the thermodynamic equilibrium composition and temperature of the produced syngas are determined.

As the target composition of the produced syngas, one can choose, e.g., a composition with the maximum hydrogen content or a composition with the maximum methane content. In the following, for the sake of definiteness, a gas containing the maximum amount of hydrogen will be referred to as syngas, while a gas containing the maximum amount of methane will be referred to as energy gas.

## 3. Results and Discussion

In this section, the properties of the GA produced by the gaseous detonations of the methane and syngas in the PDG are first demonstrated by presenting the predicted detonation parameters (detonation velocity, pressure, temperature, and density) and the composition of the detonation products of the stoichiometric methane–oxygen, methane–oxygen–steam, methane–air, and methane–oxygen-enriched air mixtures in the CJ state and in the states obtained when the CJ detonation products are expanded to pressure $P_0 \leq P \leq P_{CJ}$. Thereafter, the parameters of the syngas and energy gas predicted by models 1 and 2 are presented. Note that the products of the gaseous detonations have never been considered as a GA for organic waste gasification. In view of this, the results and analyses presented below are the novel and distinguishing features of the present research.

### 3.1. Detonation Parameters of Methane–Oxygen–Nitrogen–Steam Mixtures

### 3.1.1. Stoichiometric Methane–Oxygen Mixture

Figure 3 presents the results of the calculations for the equilibrium states of the detonation products of the stoichiometric methane–oxygen mixture: from values at the CJ state (shown by closed circles at the right) to values corresponding to the expansion of the detonation products to $P_0 = 1$ bar (shown by open squares at the left). The estimated CJ detonation velocity is $D_{CJ} = 2383$ m/s (Mach number $M_{CJ} = 6.74$). The temperature, pressure, and density of the detonation products in the CJ state are 3700 K, 29.4 bar, and 2 kg/m$^3$, respectively. The composition of the detonation products in the CJ state includes $H_2O$ (36 vol%), CO (16 vol%), $CO_2$ (10 vol%), $H_2$ (8 vol%), $O_2$ (8 vol%), as well as active radicals OH, O, and H (total 22 vol%). The temperature of the detonation products expanded to $P_0 = 1$ bar is 2852 K. The expanded detonation products include $H_2O$ (48 vol%), CO (12 vol%), $CO_2$ (17 vol%), $H_2$ (6 vol%), $O_2$ (7 vol%), as well as active radicals OH, O, and H (total 10 vol%).

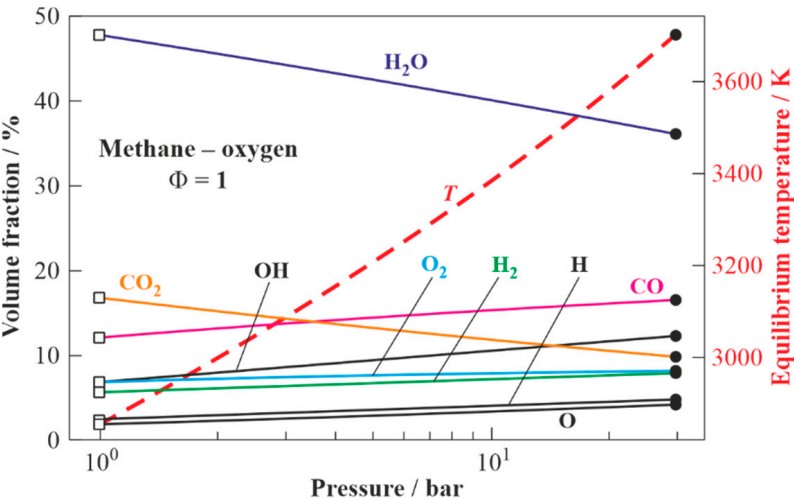

**Figure 3.** The equilibrium temperature and composition of the detonation products of the stoichiometric methane–oxygen mixture in the CJ state (shown by closed circles) and after expansion to pressure $P_0 \leq P \leq P_{CJ}$. Open squares correspond to the state at $P_0 = 1$ bar.

### 3.1.2. Stoichiometric Methane–Oxygen–Steam Mixture

Figure 4 presents the results of the calculations for the composition and temperature of the detonation products of the stoichiometric methane–oxygen–steam mixture with steam dilution ranging from 0 to 40 vol% [43] in the CJ state (Figure 4a) and after expansion to $P_0 = 1$ bar (Figure 4b). The increase in steam dilution results in the decrease in the temperature of the detonation products from 3703 to 3080 K in Figure 4a and from 2850 to 2450 K in Figure 4b, while the concentration of steam in the detonation products increases from 36 to 70 vol% in Figure 4a and from 48 to 74 vol% in Figure 4b, whereas the concentration of $CO_2$ in the detonation products remains almost constant (10–12 vol% in Figure 4a and 17–16 vol% in Figure 4b).

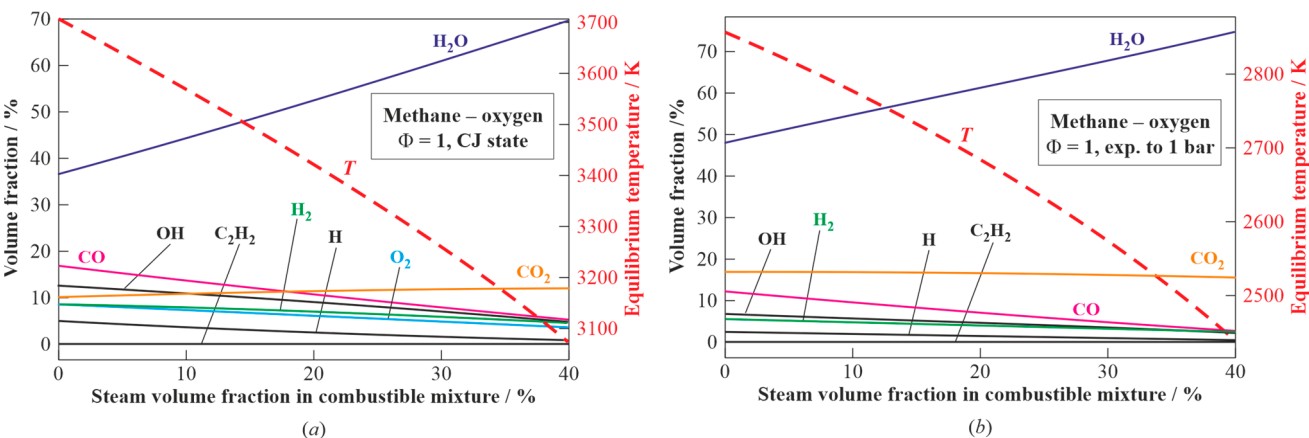

**Figure 4.** The equilibrium temperature and composition of the detonation products of the stoichiometric methane–oxygen–steam mixture in the CJ state (**a**) and after expansion to $P_0 = 1$ bar (**b**).

### 3.1.3. Stoichiometric Methane–Air Mixture

Figure 5 presents the results of the calculations for the equilibrium states of the detonation products of the stoichiometric methane–air mixture: from values in the CJ state (shown by closed circles at the right) to values corresponding to the expansion of the detonation products to $P_0 = 1$ bar (shown by open squares at the left). The estimated detonation velocity is $D_{CJ} = 1805$ m/s (Mach number $M_{CJ} = 5.14$). The temperature, pressure, and density of the detonation products in the CJ state are 2782 K, 17.3 bar, and 2 kg/m$^3$, respectively. The composition of the detonation products in the CJ state includes

$H_2O$ (17.1 vol%), CO (2.1 vol%), $CO_2$ (7.1 vol%), $O_2$ (1.2 vol%), $H_2$ (1 vol%), as well as active radicals OH, O, and H (total 1.5 vol%); the rest is nitrogen (70 vol%). The temperature of the detonation products expanded to $P_0 = 1$ bar is 1750 K. The expanded detonation products include $H_2O$ (19 vol%), $CO_2$ (9.5 vol%), and $N_2$ (71.5 vol%).

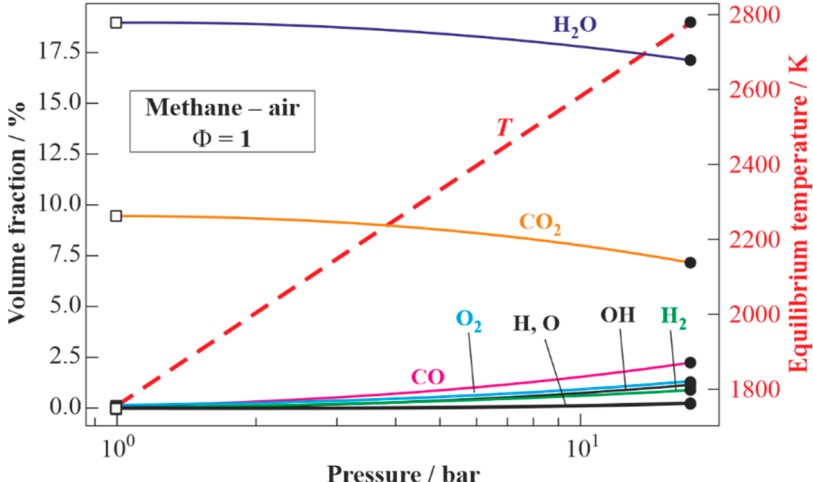

**Figure 5.** The equilibrium temperature and composition of the detonation products of the stoichiometric methane–air mixture in the CJ state (shown by closed circles) and after expansion to pressure $P_0 \leq P \leq P_{CJ}$. Open squares correspond to the state at $P_0 = 1$ bar.

3.1.4. Stoichiometric Methane—Oxygen-Enriched Air Mixture

Figure 6 shows the calculated dependences of the CJ detonation velocity (Figure 6a) and the temperature of the detonation products in the CJ state and after expansion to $P_0 = 1$ bar (Figure 6b) on the volume fraction of the oxygen in the air. When moving from the methane–oxygen to methane–air mixture, the detonation velocity $D_{CJ}$ is seen to monotonically decrease from 2383 to 1805 m/s. The temperature of the detonation products in the CJ state decreases from 3703 to 2782 K, and the temperature of the detonation products expanded to $P_0 = 1$ bar decreases from 2852 to 1750 K.

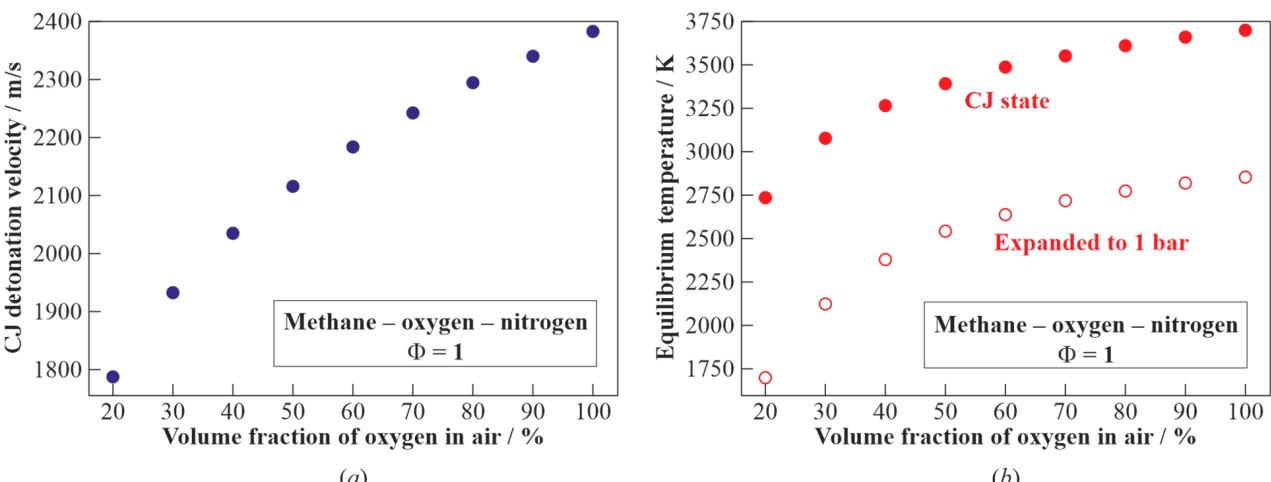

**Figure 6.** The calculated dependences of the CJ detonation velocity $D_{CJ}$ (**a**) and the temperatures of the detonation products in the CJ state and after expansion to $P_0 = 1$ bar (**b**) on the oxygen volume fraction in the air.

### 3.2. Gasification of WO by Detonation Products of the Stoichiometric Fuel–Oxygen Mixture

#### 3.2.1. Model 1

Figure 7 presents the equilibrium temperatures and compositions of the dry products of WO gasification in the detonation products of the stoichiometric methane–oxygen mixture expanded to $P_0 = 1$ bar (see Figure 3) as a function of the WO/GA mass ratio. As the target composition of dry gasification products, one can choose a composition with the maximum hydrogen content (syngas) or a composition with the maximum methane content (energy gas).

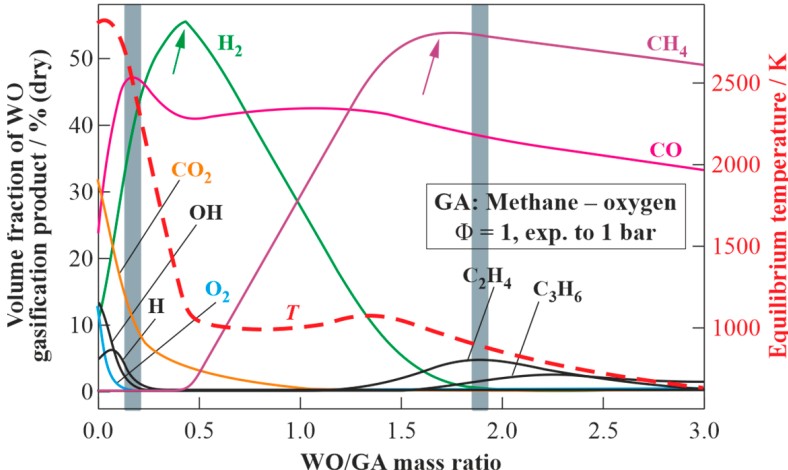

**Figure 7.** The equilibrium temperature and composition of the dry gasification products of WO as a function of the WO/GA mass ratio; GA is represented by the detonation products of the stoichiometric methane–oxygen mixture expanded to $P_0 = 1$ bar. The arrows show the compositions of syngas and energy gas. Two grey vertical bars are plotted to conditionally explain the differences between the measured and calculated compositions of the produced syngas.

In the first case, a syngas with a ratio of $H_2/CO = 1.35$ is obtained as a result of gasification by adding 0.45 kg of WO to 1 kg of GA. This syngas is characterized by the contents of $H_2$ 55.4 vol% (dry), CO 41 vol% (dry), $CO_2$ 3.4 vol% (dry), $CH_4$ 0.2 vol% (dry), temperature 1121 K, and lower heating value (LHV) 19.9 MJ/kg. If instead of the WO/GA mass ratio one considers the WO/fuel (methane) mass ratio, it turns out that with the help of 1 kg of $CH_4$ and 4 kg of $O_2$, it is feasible to gasify 2.2 kg of WO and obtain 7.2 kg of syngas with the composition specified above.

To obtain the second target composition with a methane content of 53.9 vol% (dry), it is necessary to gasify 1.73 kg of WO using 1 kg of GA. In addition to $CH_4$, such energy gas contains CO (39.3 vol% (dry)), $H_2$ (1.2 vol% (dry)), $C_2H_4$ (4 vol% (dry)), $C_2H_2$ (0.7 vol% (dry)), and $C_3H_6$ (0.8 vol% (dry)). The temperature and LHV of such energy gas are 952 K and 33.6 MJ/kg, respectively. If instead of the WO/GA mass ratio one uses the WO/fuel (methane) mass ratio, then it turns out that with the help of 1 kg of $CH_4$ and 4 kg of $O_2$ it is feasible to gasify 8.63 kg of WO and obtain 13.63 kg of energy gas with the composition specified above.

It is instructive to compare the results of the calculations in Figure 7 with experimental data for the gasification of waste machine oil with the GA obtained by the pulsed detonations of the stoichiometric natural gas—oxygen mixture [37]. Experiments in [37] were conducted with a pulsed detonation frequency of 1 Hz. The pressure in the gasifier was slightly above atmospheric pressure, i.e., $P_0 \approx 1$ bar. The liquid WO was fed to the PDG either at the open end or inside the PDG. Table 1 compares the results of the experiments with thermodynamic calculations in terms of the WO mass flow rate, $G_w$, the total mass flow rate of GA (fuel plus oxygen), $G_f + G_{ox}$, the ratio of mass flow rates of WO and GA, $G_w/\left(G_f + G_{ox}\right)$, the measured wall temperature of the gasifier, $T_w$, the calculated

equilibrium gas temperature in the gasifier, $T_g$, and the contents of $CO_2$, $CO$, $H_2$, $CH_4$, $O_2$, and $C_xH_y$ in the dry syngas (the numbers in parentheses correspond to calculations). Note that in the experiments, the local instantaneous temperatures of the GA in the gasifier exceeded 2800 K, so the gasification reactions proceeded in the wide range of temperatures between the wall temperature and GA temperature.

**Table 1.** Comparison of measured [37] and calculated results for the gasification of the waste machine oil by the detonation products of the stoichiometric methane–oxygen mixture.

| $G_w$, g/s | $G_f + G_{ox}$, g/s | $G_w/(G_f + G_{ox})$, | $T_w$, K | $T_g$, K | $CO_2$ vol% Dry | $CO$ vol% Dry | $H_2$ vol% Dry | $CH_4$ vol% Dry | $O_2$ vol% Dry | $C_xH_y$ vol% Dry |
|---|---|---|---|---|---|---|---|---|---|---|
| 2.70 | 2.65 | 1.02 | 853 | 1000 | 13 (1) | 36 (43) | 26 (28) | 13 (28) | 0.0 (0) | 11 (0) |
| 3.24 | 2.75 | 1.18 | 873 | 1000 | 9 (0) | 40 (43) | 28 (20) | 15 (35) | 0.4 (0) | 9 (0) |
| 1.90 | 1.60 | 1.18 | 823 | 1000 | 12 (0) | 38 (43) | 27 (20) | 12 (35) | 0.1 (0) | 11 (0) |
| 1.35 | 1.80 | 0.75 | 843 | 1000 | 10 (2) | 43 (42) | 31 (42) | 10 (16) | 0.0 (0) | 6 (0) |

When analyzing the data in Table 1, the following observations are worth mentioning. Firstly, the WO/GA mass ratio in the experiments [37] varied from 0.75 to 1.18, which was considerably larger than the value of WO/GA = 0.45 ensuring the maximum content of $H_2$ in the produced syngas according to Figure 7. Secondly, the measured wall temperature of the gasifier (=823–873 K) is seen to be 150–180 K less than the calculated equilibrium temperature of the gasification products (about 1000 K), which looks reasonable as the gasifier was not thermally insulated in the experiments. Thirdly, the ranges of the measured (36–43 vol% (dry)) and calculated (42–43 vol% (dry)) CO volume fractions, $H_2$ volume fractions (26–31 vol% (dry) vs. 20–42 vol% (dry)), and $O_2$ volume fractions (both approximately zero) in the produced syngas agree with some scatter but reasonably. Fourthly, the measured and calculated contents of $CO_2$, $CH_4$, and $C_xH_y$ disagree significantly: on the one hand, the calculations considerably underestimate the contents of $CO_2$ (9–13 vol% (dry) in measurements vs. 1–2 vol% (dry) in calculations) and $C_xH_y$ (6–11 vol% (dry) in measurements vs. 0 in calculations) and, on the other hand, the calculations considerably overestimate the content of $CH_4$ (10–13 vol% (dry) in measurements vs. 16–35 vol% (dry) in calculations).

Significant differences in the measured and calculated volume fractions of $CO_2$, $CH_4$, and $C_xH_y$ in the produced syngas are likely caused by the highly inhomogeneous mixing and heating of WO with GA jets in the experiments. Note that the WO/GA mass ratio in the experiments is defined as the overall rather than local mass ratio, whereas in the thermodynamic calculations, both overall and local WO/GA mass ratios are the same. Since chemical reactions proceed at the local WO/GA mass ratio, one can conditionally imply that one part of WO is gasified at the local WO/GA mass ratio less than 0.45, whereas another part of WO is gasified at the local WO/GA mass ratio higher than 0.45. This implication is conditionally illustrated in Figure 7 by two grey vertical bands. Obviously, the measured values of the $CO_2$, $CO$, $H_2$, $CH_4$, $O_2$, and $C_xH_y$ volume fractions in Table 1 can be obtained by averaging two different syngas compositions corresponding to these two conditional values of the local WO/GA mass ratio. As a matter of fact, Figure 8 shows the schematic of the shock-induced fragmentation of four WO drops. Clearly, the local WO/GA mass ratio is different for the dense cores in the mist clouds behind the drops and for the loose zones near the edges of the clouds.

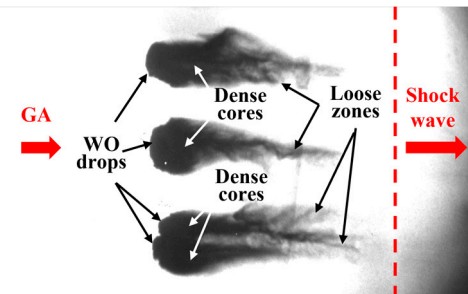

**Figure 8.** The schematic of the shock-induced fragmentation of four WO drops illustrating differences in the local WO/GA mass ratio for the dense cores in the mist clouds behind the drops and for the loose zones near the edges of the clouds.

### 3.2.2. Model 2

Model 2 implies the self-feeding of the PDG with the produced syngas. The self-feeding of the PDG is preceded by its operation on the starting fuel (methane). The composition of the syngas obtained by the detonation of the stoichiometric methane–oxygen mixture can be determined from Figure 7. To determine the composition of the syngas obtained with PDG self-feeding, it is necessary to gradually replace the starting fuel (methane) in the PDG with dry syngas obtained as a result of WO gasification. Table 2 shows the calculated parameters of the WO gasification process with one cycle (cycle 0) with the starting fuel (methane) and ten subsequent cycles (cycles 1 to 10) with PDG self-feeding with the stoichiometric mixture of the produced syngas with oxygen. Firstly, the detonation velocity $D_{CJ}$ of the stoichiometric syngas–oxygen mixture exceeds 2082 m/s, which is only 14.5% less than the detonation velocity of the starting fuel ($D_{CJ}$ = 2383 m/s). Secondly, by already the 8th–9th cycle the composition and temperature of the produced syngas are established: the addition of 0.37 kg of WO to 1 kg of GA results in the production of a syngas with a ratio of $H_2$/CO = 1.04, with a temperature of 1095 K, with a high content of $H_2$ (48.4 vol% (dry)) and CO (46.4 vol% (dry)), with a low content of $CO_2$ (4.8 vol% (dry)), and with negligible contents of $C_2$–$C_3$ hydrocarbons. The LHV of such syngas is 17.9 MJ/kg. Using 1 kg of such syngas and 1.28 kg of $O_2$, it is feasible to gasify 0.81 kg of WO and obtain 3.09 kg of the identical syngas, i.e., the conversion of 1 kg of WO in the GA requires 32.4% of the produced syngas.

**Table 2.** Parameters of the WO gasification process with PDG self-feeding with the stoichiometric syngas–oxygen mixture according to model 2.

| Cycle No. | $D_{CJ}$ m/s | Fuel | $H_2$ vol% Dry | CO vol% Dry | $CO_2$ vol% Dry | $O_2$ vol% Dry | $CH_4$ vol% Dry | $C_2H_2$ vol% Dry | $C_2H_4$ vol% Dry | $C_3H_6$ vol% Dry | $T,$ K | WO/GA | LHV, MJ/kg |
|---|---|---|---|---|---|---|---|---|---|---|---|---|---|
| 0 | 2383 | $CH_4$ | 0.554 | 0.410 | 0.034 | 0 | 0.003 | 0 | 0 | 0 | 1121 | 0.45 | 49.644 |
| 1 | 2158 | SG | 0.551 | 0.443 | 0.042 | 0 | 0.004 | 0 | 0 | 0 | 1096 | 0.40 | 19.870 |
| 2 | 2112 | SG | 0.495 | 0.456 | 0.046 | 0 | 0.003 | 0 | 0 | 0 | 1100 | 0.38 | 18.642 |
| 3 | 2093 | SG | 0.488 | 0.461 | 0.048 | 0 | 0.002 | 0 | 0 | 0 | 1110 | 0.37 | 18.135 |
| 4 | 2084 | SG | 0.485 | 0.463 | 0.049 | 0 | 0.003 | 0 | 0 | 0 | 1098 | 0.37 | 17.901 |
| 5 | 2082 | SG | 0.484 | 0.464 | 0.048 | 0 | 0.003 | 0 | 0 | 0 | 1096 | 0.37 | 17.876 |
| 6 | 2082 | SG | 0.484 | 0.464 | 0.048 | 0 | 0.003 | 0 | 0 | 0 | 1095 | 0.37 | 17.866 |
| 7 | 2082 | SG | 0.484 | 0.464 | 0.048 | 0 | 0.003 | 0 | 0 | 0 | 1095 | 0.37 | 17.863 |
| 8 | 2082 | SG | 0.484 | 0.464 | 0.048 | 0 | 0.003 | 0 | 0 | 0 | 1095 | 0.37 | 17.861 |
| 9 | 2082 | SG | 0.484 | 0.464 | 0.048 | 0 | 0.003 | 0 | 0 | 0 | 1095 | 0.37 | 17.861 |
| 10 | 2082 | SG | 0.484 | 0.464 | 0.048 | 0 | 0.003 | 0 | 0 | 0 | 1095 | 0.37 | 17.861 |

SG stands for syngas.

### 3.3. Gasification of WO by Detonation Products of the Stoichiometric Fuel–Air Mixture

3.3.1. Model 1

It is interesting to determine the expected composition of syngas when replacing oxygen with air in the PDG. In this case, the composition and temperature of the GA for model 1 are determined from Figure 5 (values at = 1 bar). Figure 9 presents the results of the calculations of the equilibrium states of the WO gasification products as a function of the WO/GA mass ratio. As the target composition of the dry gasification products, one can choose, e.g., a composition with the maximum hydrogen content (syngas) or a composition with the maximum methane content (energy gas). In the first case, when adding 0.09 kg of WO to 1 kg of GA, the gasification process results in a syngas with a ratio of $H_2/CO = 1.7$ with the contents of $H_2$ 23.7 vol% (dry), CO 13.8 vol% (dry), $CO_2$ 6.0 vol% (dry), $N_2$ 55.5 vol% (dry), and $CH_4$ 1.0 vol% (dry) with a temperature of 928 K and an LHV of 4.6 MJ/kg. If instead of the WO/GA mass ratio one uses the WO/fuel (methane) mass ratio, it turns out that with the help of 1 kg of $CH_4$ and 17.2 kg of air it is feasible to gasify 1.65 kg of WO and obtain 19.8 kg of syngas with the specified composition. The calculated detonation velocity of the stoichiometric syngas–air mixture is $D_{CJ} = 1802$ m/s, which is at the level of values realized in practice for the fuel–air mixtures of a number of hydrocarbon fuels. In the second case, using 1 kg of GA, it is possible to gasify 0.42 kg of WO to produce an energy gas containing $CH_4$ (32.0 vol% (dry)), CO (23.3 vol% (dry)), $C_2H_4$ (0.3 vol% (dry)), and $N_2$ (44.4 vol% (dry)) with a temperature of 763 K and an LHV of 13.6 MJ/kg.

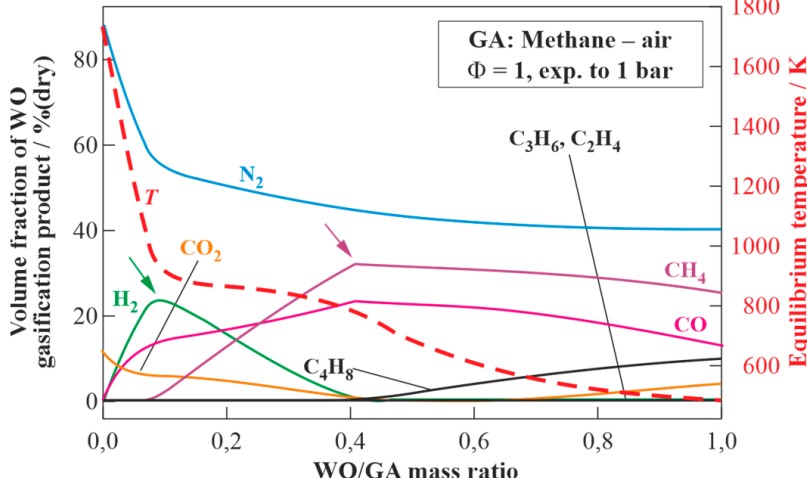

**Figure 9.** The equilibrium temperature and composition of the dry gasification products of WO as a function of the WO/GA mass ratio; GA is represented by the detonation products of the stoichiometric methane–air mixture expanded to $P_0 = 1$ bar. The arrows show the compositions of syngas and energy gas.

3.3.2. Model 2

Model 2 implies the self-feeding of the PDG with the produced syngas. The self-feeding is preceded by PDG operation on starting fuel (methane). The composition of the syngas obtained by the detonation of the stoichiometric methane–air mixture can be determined from Figure 9. To determine the composition of the syngas obtained with PDG self-feeding, it is necessary to gradually replace the starting fuel (methane) in the PDG with dry syngas produced by WO gasification. These manipulations can be accomplished by gradually replacing methane with the syngas of the resulting composition. Calculations show that the temperature and composition of the produced syngas are established only by the 14th–15th cycle: when adding 0.03 kg of WO to 1 kg of GA, gasification results in a syngas with a ratio of $H_2/CO = 1.76$ and with a temperature of 785 K (the contents of $H_2$ 7.1 vol% (dry), CO 4.0 vol% (dry), $CO_2$ 12.9 vol% (dry), $CH_4$ 1.7 vol% (dry), and $N_2$

74.3 vol% (dry)). The LHV of such syngas is 1.5 MJ/kg, and the calculated detonation velocity of the stoichiometric syngas–air mixture is below 1300 m/s, which is over 40% less than the detonation velocity of the starting fuel (=1805 m/s). Since such a detonation velocity looks too low for organizing a reliable detonation process, one can expect that the gasification process according to model 2 is not feasible in practice.

### 3.4. Gasification of WO by Detonation Products of the Stoichiometric Fuel–Oxygen-Enriched Air Mixture

#### 3.4.1. Model 1

Figure 10 presents the results of the calculations of the equilibrium states of the dry products of WO gasification using GA, obtained by the detonation of the stoichiometric methane–oxygen–nitrogen mixture with different concentrations of oxygen in the air, but at a fixed WO/GA mass ratio of 0.45 (see Figure 7). With increasing oxygen concentration in the air, the contents of $H_2$, CO, and $CO_2$ in the produced syngas monotonically increase, and the content of $CH_4$ monotonically decreases (at $O_2$ concentration in the air above 25 vol% (dry)). As the target composition of the dry gasification products, one can choose, e.g., a composition with the maximum content of $H_2$ (syngas) or a composition with the maximum content of $CH_4$ (energy gas). In Figure 9, these compositions are shown by arrows. In the first case, the maximum volume fraction of $H_2$ (55.2 vol% (dry)) is achieved when pure oxygen is used as GA. The LHV of such syngas is 19.9 MJ/kg. An energy gas with the maximum volume fraction of $CH_4$ (33.6 vol% (dry)) is obtained with the content of $O_2$ in the air of 23 vol% (dry). The LHV of such energy gas is 17.9 MJ/kg.

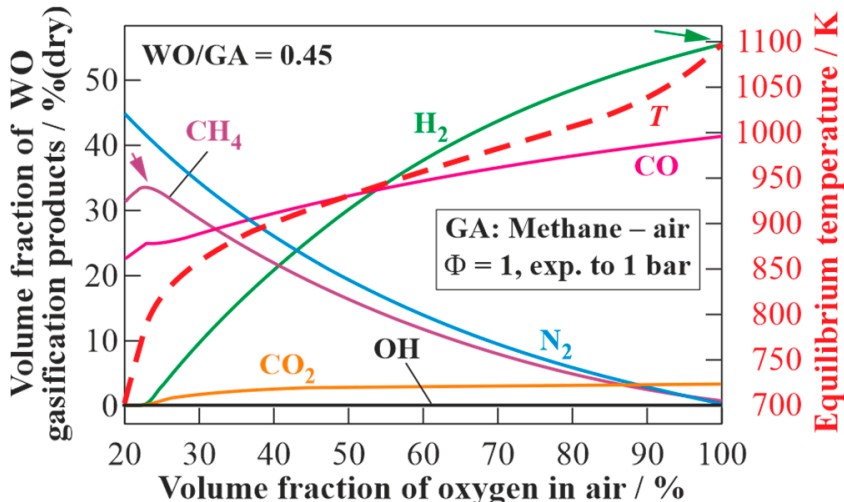

**Figure 10.** The equilibrium temperature and composition of the dry gasification products of WO as a function of the oxygen concentration in the air at a fixed WO/GA mass ratio of 0.45; GA is represented by the detonation products of the stoichiometric methane–oxygen-enriched air mixture expanded to $P_0 = 1$ bar. The arrows show the compositions of syngas and energy gas.

#### 3.4.2. Model 2

Figure 11 presents the results of the calculations of the equilibrium states of the dry gasification products of WO using GA, obtained by the detonation of the stoichiometric syngas ($H_2$/CO = 1.3)–oxygen–nitrogen mixture with different oxygen concentrations in the air, but at a fixed WO/GA mass ratio of 0.45 (see Figure 7). With increasing oxygen concentration in the air, the contents of $H_2$ and CO in the produced syngas monotonically increase, and the content of $CH_4$ monotonically decreases. As the target composition of the dry gasification products, one can choose, e.g., a composition with the maximum content of $H_2$ (syngas) or a composition with the maximum content of $CH_4$ (energy gas). In the first case, the maximum content of $H_2$ (51 vol% (dry)) is achieved when using pure oxygen as GA. The LHV of such syngas is 16.7 MJ/kg. In the second case, an energy gas with the

maximum content of $CH_4$ (34 vol% (dry)) is obtained with an oxygen concentration in the air of 21 vol% (dry). The LHV of such energy gas is 20.3 MJ/kg.

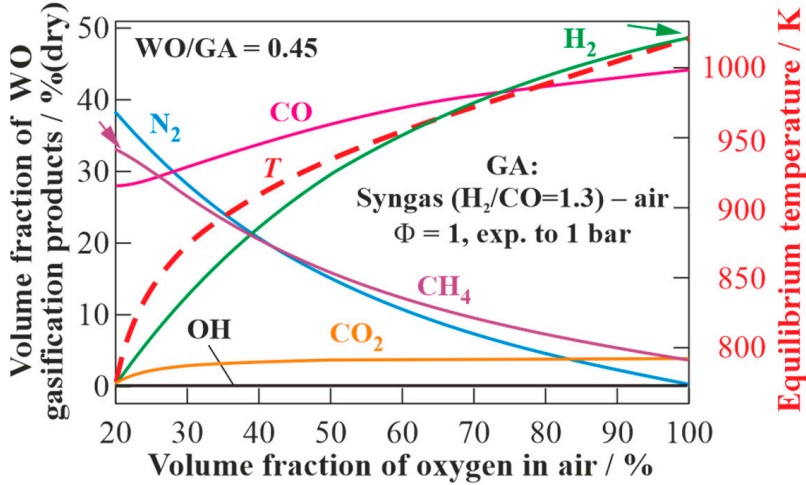

**Figure 11.** The equilibrium temperature and composition of the dry gasification products of WO as a function of the oxygen concentration in the air at a fixed WO/GA mass ratio of 0.45; GA is represented by the detonation products of the stoichiometric syngas ($H_2$/CO = 1.3)—oxygen-enriched air mixture expanded to $P_0$ = 1 bar. The arrows show the compositions of syngas and energy gas.

*3.5. Simple Estimates*

Thus, when using methane as fuel and oxygen as oxidizer for PDG and operating the gasifier at $P_0$ = 1 bar, calculations with model 1 show that using 1 kg of $CH_4$ and 4 kg of $O_2$, it is feasible to gasify 2.2 kg of WO and obtain 7.2 kg of syngas with an $H_2$ content of 55.4 vol% (dry) and CO content of 41 vol% (dry) (with an $H_2$/CO ratio of 1.35) and an LHV of 19.9 MJ/kg. If the resulting syngas is used as fuel for the PDG according to model 2, then the portion of the syngas for PDG self-feeding will be 32.4%, i.e., approximately one-third, and the remaining two-thirds of the produced syngas can be delivered to a customer. Table 3 shows the calculated $H_2$/CO ratio and composition of the dry syngas obtained by model 2.

**Table 3.** The equilibrium composition of the dry syngas obtained by WO gasification at $P_0$ = 1 bar with the detonation products of the stoichiometric fuel–oxygen mixture, calculated using model 2 and Equation (1).

| Source | PDG | | Syngas | | | | |
|---|---|---|---|---|---|---|---|
| | Fuel | Oxidizer | $H_2$/CO | $H_2$ vol% Dry | Co Vol% Dry | $CO_2$ vol% Dry | $CH_4$ vol% Dry |
| Model 2 | $CH_4$/Syngas | $O_2$ | 1.04 | 48.4 | 46.4 | 4.8 | 0.3 |
| Equation (1) | $CH_4$/Syngas | $O_2$ | 1.06 | 51.5 | 48.5 | 0 | 0 |

The results of the thermodynamic calculations presented in Table 3 can be compared with simple estimates based on the overall reaction of the high-temperature gasification of $C_{16}H_{34}$ by steam and carbon dioxide with the complete conversion of carbon into CO:

$$C_{16}H_{34} + a[16CO_2 + 17H_2O] = b[16CO + 17H_2] \tag{1}$$

It is easy to show that the stoichiometric coefficients of such a reaction are $a$ = 0.485 and $b$ = 1.485. It follows from Equation (1) that with the help of 1 kg of GA it is feasible to gasify 0.46 kg of WO and obtain a syngas with a ratio of $H_2$/CO = 1.06 with a hydrogen content of 51.5 vol% (dry) and CO content of 48.5 vol% (dry). These results are compared with the results of the thermodynamic calculations in the last line of Table 3. The values

of the $H_2$/CO ratio and syngas composition obtained from Equation (1) are in reasonable agreement with the thermodynamic calculations using model 2. Equation (1) slightly overestimates the quality of the produced syngas. The reason for this overestimation is that Equation (1) assumes the complete conversion of carbon into CO, which implies a gasification temperature exceeding 1500 K. From now on, the term "gasification temperature" implies the equilibrium temperature of the gasification products. However, the thermodynamic calculations show a relatively low gasification temperature of 1095 K and the incomplete conversion of carbon into CO: the syngas in model 2 contains 4.8 vol% (dry) $CO_2$ and 0.3 vol% (dry) $CH_4$.

To enhance the carbon conversion efficiency, one can increase the gasification temperature. The gasification temperature in models 1 and 2 can be increased by increasing the gasification pressure $P$ in the gasifier. This can be done by making a provision for a pressure relief valve at the gasifier outlet (see Figure 2). The valve could vent the gasifier after each detonation pulse when the pressure exceeds a certain preset value. In this case, the CJ detonation products will expand to the gasifier with the growing backpressure. Thus, according to Figure 3, when the gasification process of WO in model 1 is carried out at an intermediate pressure $P_0 \leq P \leq P_{CJ}$, the gasification temperature increases from 2852 to 3700 K. However, on the one hand, an increase in the gasification pressure $P$ leads to the variation in GA composition: the concentrations of $H_2O$ and $CO_2$ in the GA decrease from 48 to 36 vol% and from 17 to 10 vol%, respectively, whereas the concentrations of CO and $H_2$ increase from 12 to 17 vol% and from 6 to 8 vol%, respectively. As for the O, H, and OH radicals and molecular oxygen $O_2$, their concentrations increase from 10 to 21 vol% and from 7 to 8 vol%, respectively. On the other hand, the composition of the GA can be controlled by diluting the combustible mixture in the PDG with low-temperature steam. Thus, according to Figure 4, with an increase in the steam dilution degree of the stoichiometric methane–oxygen mixture from 0 to 40 vol%, the temperature of the detonation products in model 1 decreases from 2850 to 2450 K, but the content of steam in the GA in model 1 increases from 48 to 74 vol%, whereas the content of $CO_2$ remains approximately at the same level of 17–16 vol%, respectively. The addition of steam to the stoichiometric methane–oxygen mixture in model 1 increases the content of $H_2$ in the produced syngas from 55.4 to 62.8 vol% (dry) and reduces the content of CO in model 1 from 41.2 to 18.6 vol% (dry), thus changing the $H_2$/CO ratio in the resulting syngas from 1.3 to 3.4.

Let us now estimate the amount of the produced syngas utilized for PDG self-feeding. To produce the GA entering the left side of Equation (1), it is necessary to spend some amounts of the produced syngas and oxygen, which can be determined from the following reaction equation:

$$[16CO + 17H_2] + 16.5O_2 = 16CO_2 + 17H_2O. \tag{2}$$

It follows from Equation (2) that for producing 1 kg of GA it is necessary to consume 0.48 kg of syngas and 0.52 kg of oxygen. Using the results of the analysis of Equation (1), one finds that for the gasification of 0.46 kg of WO it is necessary to consume 0.48 kg of syngas and 0.52 kg of oxygen. As a result of gasification, $0.46 + 0.48 + 0.52 = 1.46$ kg of syngas is obtained, i.e., 32.8% of the produced syngas is used for PDG self-feeding. This result is in good agreement with the thermodynamic calculation for model 2, in which 32.4% of the produced syngas is consumed in PDG self-feeding. This means that the approach presented herein can be used for the simple estimates of self-feeding needs.

If one considers energy gas as the target product of WO gasification according to model 1, then with the help of 1 kg of $CH_4$ and 4 kg of $O_2$ it is feasible to gasify 8.63 of WO and obtain 13.63 kg of energy gas with a content of $CH_4$ of 53.9 vol% (dry), CO 39.3 vol% (dry), $H_2$ 1.2 vol% (dry), $C_2H_4$ 4 vol% (dry), $C_2H_2$ 0.7 vol% (dry), $C_3H_6$ 0.8 vol% (dry), etc. The LHV of such energy gas reaches 33.7 MJ/kg.

When using methane as fuel and air as oxidizer for PDG, calculations with model 1 show that using 1 kg of methane and 17.2 kg of air, it is feasible to gasify 1.65 kg of WO and obtain 19.8 kg of syngas with the contents of $H_2$ 23.7 vol% (dry) and CO 13.8 vol% (dry) with $H_2$/CO = 1.7, an LHV of 4.6 MJ/kg, and a gasification temperature of 928 K.

By increasing the backpressure in the gasifier and enriching the air with oxygen, it looks possible to implement in practice the process of WO gasification according to model 2. On the one hand, according to Figure 5, when operating a PDG on the stoichiometric methane–air mixture at $P_0$ = 1 bar, the increase in the gasification pressure $P$ from 1 to 5 bar will lead to an increase in the gasification temperature from 1750 to 2300 K. On the other hand, according to Figure 6, the enrichment of air with oxygen leads to the monotonic increase in the detonation velocity of the stoichiometric methane–oxygen–nitrogen mixture from 1805 to 2382 m/s and an increase in the CJ temperature of the detonation products from 2782 to 3700 K. With an increase in the backpressure in the gasifier and/or oxygen content in the air, the content of $H_2$ and CO in the produced syngas in model 1 monotonically increases. It can be expected that at certain values of backpressure in the gasifier and/or oxygen content in the air, WO gasification with PDG self-feeding with the produced syngas will become practically feasible.

Let us now estimate the amount of produced syngas that can be used for the self-feeding of the PDG operating on the fuel–air mixture implying the complete conversion of carbon into CO. In this case, the overall reactions of Equations (1) and (2) will take the following form:

$$C_{16}H_{34} + 0.485[16CO_2 + 17H_2O + 92.14N_2] = 1.485[16CO + 17H_2 + 30.1N_2]; \qquad (3)$$

$$[16CO + 17H_2 + 30.1N_2] + [16.5O_2 + 62.04N_2] = 16CO_2 + 17H_2O + 92.14N_2. \qquad (4)$$

Firstly, it follows from Equation (3) that the gasification of 1 kg of WO with 7.7 kg of GA will produce 8.7 kg of syngas diluted with nitrogen. The syngas will have $H_2$/CO = 1.06 and the contents of $H_2$, CO, and $N_2$ will be 26.9, 25.4, and 47.7 vol% (dry), respectively. Secondly, it follows from Equations (3) and (4) that for producing 1 kg of GA one needs to consume 0.369 kg of syngas and 0.631 kg of air, i.e., $100 \times 7.7 \times 0.369/8.7$ = 32.7% is consumed on PDG self-feeding with the produced syngas. This result agrees satisfactorily with the results of the thermodynamic calculations using model 2 and the estimates based on Equations (1) and (2) for the case when the PDG operates on the fuel–oxygen mixture (32.4% and 32.8%, respectively).

If one considers energy gas as the target product of WO gasification by the detonation products of the stoichiometric methane–air mixture at $P_0$ = 1 bar, then according to model 1 with the help of 1 kg of GA it is feasible to gasify 0.45 kg of WO and obtain an energy gas with the contents of $CH_4$ 32 vol% (dry), CO 23.3 vol% (dry), $N_2$ 44.4 vol% (dry), and $C_2H_4$ 0.3 vol% (dry) with a gasification temperature of 763 K and an LHV of 13.6 MJ/kg.

## 4. Conclusions

The most important results obtained herein are summarized below:

- The use of the detonation products of the stoichiometric methane–oxygen and methane–air mixtures for the gasification of waste oil theoretically allows achieving the complete conversion of waste oil into a syngas consisting exclusively of $H_2$ and CO, or into an energy gas with high contents of $CH_4$ and $C_2$-$C_3$ hydrocarbons and an LHV of 36.7 (fuel–oxygen mixture) and 13.6 MJ/kg (fuel–air mixture).
- The use of the detonation products of the stoichiometric mixture of the produced syngas with oxygen or with oxygen-enriched air for oil gasification also theoretically allows achieving the complete conversion of waste oil into a syngas consisting exclusively of $H_2$ and CO.
- About 33% of the produced syngas mixed with oxygen can be theoretically used for the self-feeding of the pulsed detonation gun. To self-feed the pulsed detonation gun with a mixture of the produced syngas with air, it is necessary to increase the backpressure in the gasifier and/or enrich the air with oxygen. The self-feeding of the pulsed detonation gun with the produced syngas and the replacement of oxygen by oxygen-enriched air make the gasification technology more attractive and cost-effective.

- The addition of low-temperature steam to the fuel–oxygen mixture in the pulsed detonation gun allows controlling the composition of the produced syngas within a wide range. Theoretically, the $H_2/CO$ ratio can vary from 1.3 to 3.4. Depending on the $H_2/CO$ ratio, the syngas can be suitable for different applications. $H_2$-rich syngas with a large $H_2/CO$ ratio can be used for producing pure $H_2$ or for the synthesis of $NH_3$. Syngas with an $H_2/CO$ ratio ranging from 1 to 2 is suitable for producing transportation fuels and methanol.
- To approach the theoretical compositions of the produced syngas or energy gas in experiments, one must pay special attention to the homogeneous mixing of organic feedstock with the gasification agent.

It is worth emphasizing that thermodynamic modeling generally provides the trends rather than the actual values of temperature and product composition. The differences between calculations and experiments are usually attributed to the imperfect mixing of components, the finite rates of heat and mass transfer and chemical transformations, as well as thermal losses.

Future work will be focused on the experimental implementation of the new technology of organic waste gasification with the self-feeding of the pulsed detonation gun with the produced syngas. Further efforts will also be made to include the various other properties of organic wastes in the thermodynamic analysis, like the water, oxygen, and chlorine content.

**Author Contributions:** Conceptualization, S.M.F.; methodology, S.M.F. and V.A.S.; software, V.A.S. and K.S.P.; validation, V.A.S. and K.S.P.; formal analysis, S.M.F. and V.A.S.; investigation, V.A.S. and K.S.P.; resources, S.M.F.; data curation, K.S.P.; writing—original draft preparation, S.M.F.; writing—review and editing, S.M.F.; supervision, S.M.F.; project administration, S.M.F.; funding acquisition, S.M.F. All authors have read and agreed to the published version of the manuscript.

**Funding:** This research received no external funding.

**Data Availability Statement:** Data will be available on request.

**Conflicts of Interest:** The authors declare no conflicts of interest.

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
