# Peer review of "Gasification of Liquid Hydrocarbon Waste by the Ultra-Superheated Mixture of Steam and Carbon Dioxide: A Thermodynamic Study"

_energies, doi:10.3390/en17092126_

Round 1

Reviewer 1 Report

Comments and Suggestions for Authors

The article presents adequate scientific content. However, there are points for improvement:

The introduction must contain the complete and updated state of the art, showing the article's novelty. The methodology should be written in greater detail, making it easy for the reader to understand.

The results and discussion must be improved to highlight the differences concerning the literature.

Comments on the Quality of English Language

In general, the writing should be improved extensively, avoiding phrases such as "it is explained as follows, "and "if we consider."

Author Response

We are grateful to the reviewer for valuable comments. We have made our best to follow all the comments. All changes in the revised manuscript are marked in yellow.

The article presents adequate scientific content. However, there are points for improvement:

The introduction must contain the complete and updated state of the art, showing the article's novelty.

To follow this comment, we have added several paragraphs with the noteworthy contributions in thermodynamic simulation of low-temperature and high-temperature H2O/CO2-assisted gasification. In view of it, some references in the manuscript are renumbered.

The methodology should be written in greater detail, making it easy for the reader to understand.

To follow this comment, we have added a new figure with a sketch of gasification plant (Fig. 1). In view of it, all other figures in the manuscript are renumbered.

The results and discussion must be improved to highlight the differences concerning the literature.

To follow this comment, we have added a couple of sentences at the beginning of Section 3. Results:

“Note that the products of gaseous detonations have never been considered as a GA for organic waste gasification. In view of it, the results and analysis presented below are the novel and distinguishing features of the present research.”

Comments on the Quality of English Language

In general, the writing should be improved extensively, avoiding phrases such as "it is explained as follows, "and "if we consider."

We have made our best to avoid first person narration, thank you.

Reviewer 2 Report

Comments and Suggestions for Authors

The manuscript covers an interesting topic, but it also contains a number of weak points listed below that need to be improved. It can be considered for publication after major revisions.

1. General: The major weak point of this study are differences in the modeling by approaching the gasification reactor by an adiabatic reactor, but in a real gasification reactor there are huge internal and external heat flows. It can be assumed that this is the main reason why the results of this study (gas composition, gasification temperatures etc.) clearly differ from the experiment. Why was this approach chosen? This must be added to the discussion and conclusions that this is the weak point of the used model.

2. Introduction: Line 33-44: Please add references for all benefits. L. 58: This is the gasification temperature and not GA temperature. L. 68-81: Please add more references here. General: Please perform a discussion of the gasification process in comparison to the conventional treatment (cracking etc.) of higher oil components.

3. Materials and Methods: L. 91-113: Are there any comparable studies for WO treatment by gasification in the literature?

4. Results: L. 206: Why nitrogen in this heading? Fig. 2: Why is the pressure so low compared to real gasification pressures of 65 bar and higher? Table 1: The gasification temperature is too low for a real gasification as the Boudouard reaction is not fully on the side of CO. Fig. 10: Where is the oxygen (WO/GA=0.45) going during the gasification?

5. Discussions: L. 468: The syngas has an H2/CO ratio of 1.06. What is the targeted product to synthesize from the syngas and what is the targeted H2/CO ratio in comparison to the conventional treatment?

6. Writing: L. 78: Write ammonia or NH3 instead of both. L. 228: A space is missing.  

Author Response

We are grateful to the reviewer for valuable comments. We have made our best to follow all the comments. All changes in the revised manuscript are marked in green.

The manuscript covers an interesting topic, but it also contains a number of weak points listed below that need to be improved. It can be considered for publication after major revisions.

General: The major weak point of this study are differences in the modeling by approaching the gasification reactor by an adiabatic reactor, but in a real gasification reactor there are huge internal and external heat flows. It can be assumed that this is the main reason why the results of this study (gas composition, gasification temperatures etc.) clearly differ from the experiment. Why was this approach chosen? This must be added to the discussion and conclusions that this is the weak point of the used model.

To follow this comment, we have added a paragraph at the end of Inroduction and in the Conclusions:

“It is worth emphasizing that thermodynamic modeling generally provides the trends rather than actual values of temperature and product composition. The differences between calculations and experiments are usually attributed to imperfect mixing of components, finite rates of heat and mass transfer and chemical transformations, as well as thermal losses.”

  1. Introduction:

Line 33-44: Please add references for all benefits.

We have added three more references:

Shahbeig, H.; Shafizadeh, A.; Rosen, M.A.; Sels, B.F. Exergy sustainability analysis of biomass gasification: A critical review. Biofuel Research J. 2022, 33, 1592–1607. doi: 10.18331/BRJ2022.9.1.5.

Filippova, S.P.; Keiko, A.V. Coal gasification: At the crossroads. Economic outlook. Therm. Eng. 2021, 68(5), 347–360. doi: 10.1134/S0040601521050049.

Wang, K.; Kong, G.; Zhang, G.; Zhang, X.; Han, L.; Zhang, X. Steam gasification of torrefied/carbonized wheat straw for H2-enriched syngas production and tar reduction. Int. J. Env. Res. Pub. He. 2022, 19, 10475. doi: 10.3390/ijerph191710475.

  1. 58: This is the gasification temperature and not GA temperature.

GA is deleted.

  1. 68-81: Please add more references here. General: Please perform a discussion of the gasification process in comparison to the conventional treatment (cracking etc.) of higher oil components.

To address this comment, we have added some new references and the following statement of the limitations of the existing high-temperature gasification technologies:

“In addition to these advantages, existing high-temperature plasma and solar-assisted technologies have certain limitations. Despite the typical operation temperature of plasma gasifiers is below 2000 °C, plasma technologies require large amounts of electricity for gas–plasma transition. Moreover, plasma gasifiers need special materials with a refractory lining and water cooling, as well as short-service life arc electrodes. The main limitation of solar-assisted gasifiers is their intermittent nature.”

  1. Materials and Methods: L. 91-113: Are there any comparable studies for WO treatment by gasification in the literature?

There are only several studies of WO treatment in the literature, including our experimental work [39]. In [39], we compare our measurements with some available results on low-temperature gasification of fresh synthetic engine oil with steam and supercritical water at a temperature ranging from 500 to 800 â—¦C and a high pressure ranging from 50 to 500 bar. To address this comment, we added two more references ([40,41]) to the manuscript at the end of Section 2.1.

  1. Results:
  2. 206: Why nitrogen in this heading?

Nitrogen is present because we consider the possibility of using oxygen-enriched air as an oxidizer instead of oxygen or air (see Figure 6).

Fig. 2: Why is the pressure so low compared to real gasification pressures of 65 bar and higher?

This is one of the main advantages of the technology under study. There is no need to use high pressures once the gasification temperature is above 2000 C. To address this comment, we have added some sentences on the effect of gasification pressure at the beginning of Introduction.

Table 1: The gasification temperature is too low for a real gasification as the Boudouard reaction is not fully on the side of CO.

The actual process temperature is considerably higher than the values in Table 1. As shown by the calculations, the temperature of the gasification agent entering the gasifier exceeds 2800 K. To avoid misunderstanding, we have added the following sentence before Table 1:

“Note that in the experiments, the local instantaneous temperatures of the GA in the gasifier exceeded 2800 K, so that the gasification reactions proceeded in the wide range of temperatures between the wall temperature and GA temperature.”

Fig. 10: Where is the oxygen (WO/GA=0.45) going during the gasification?

The former Figure 10 (now, Fig. 11) relates to the use of oxygen-enriched air instead of pure oxygen in the Pulsed Detonation Gun for generating the gasification agent (GA) composed mostly of steam and carbon dioxide. This GA is then used for waste oil gasification. The content of oxygen in the GA is negligible, so it does not play any role. This figure only shows that oxygen enrichment of air results in the monotonous increase of the process temperature as well as H2 and CO concentrations in the product gas.

  1. Discussions: L. 468: The syngas has an H2/CO ratio of 1.06. What is the targeted product to synthesize from the syngas and what is the targeted H2/CO ratio in comparison to the conventional treatment?

To address this comment, we have added a sentence on the possible target products to the Conclusions:

“Depending on the H2/CO ratio, the syngas can be suitable for different applications. H2-rich syngas with a large H2/CO ratio can be used for producing pure H2 or for synthesis of NH3. Syngas with the H2/CO ratio ranging from 1 to 2 is suitable for producing transportation fuels and methanol.”

  1. Writing:
  2. 78: Write ammonia or NH3 instead of both.

Done.

  1. 228: A space is missing.  

Done.

Reviewer 3 Report

Comments and Suggestions for Authors

The paper is easy to understand but some improvements can be made to show its practical indications/values. I do believe that even simulation studies should be based on facts. Please see my comments below and improve.

1.     “ultra-superheated H2O/CO2 mixture with a temperature above 1500 °C” I understand that this is a paper based on simulation. But for such a high temperature, what kind of material can be used to build the reactor? Even for simulation work, it is better to have a practical basis. It will be good to provide some background information on this…

2.     Title: “liquid hydrocarbon waste”, Section 2: “Organic waste”. Figure 1, it is hard to tell what is the waste exactly. Please indicate what is the waste…I believe the composition of the waste have a significant impact on the gasification outcome. For instance, if the waste is completely organic, can it still produce residues after gasification?

3.      Section 3 and 4, please add more analysis of the results. For example, discuss the impact of the results rather than just display the data. In fact, Results and Discussion can be combined. Please reduce the description of the data. Instead, provide some discussion based critical thoughts. For example, how those trend from different models will impact future applications, etc.

4.     Equation (1). Is it Ok to use “C16H34” to represent any organic waste? A chemical with such a composition can be used as a fuel. Also, in life, it is difficult to find organic waste that does not contain any oxygen.

5.     Conclusion. Please combine the bullets.

Author Response

We are grateful to the reviewer for valuable comments. We have made our best to follow all the comments. All changes in the revised manuscript are marked in blue.

The paper is easy to understand but some improvements can be made to show its practical indications/values. I do believe that even simulation studies should be based on facts. Please see my comments below and improve.

1.“ultra-superheated H2O/CO2 mixture with a temperature above 1500 °C” I understand that this is a paper based on simulation. But for such a high temperature, what kind of material can be used to build the reactor? Even for simulation work, it is better to have a practical basis. It will be good to provide some background information on this…

To address this comment, we have added the following sentences to Section 2.1:

“In general, the operation process of the PDG–gasifier assembly has much in common with reciprocating engines. The gasifier can be made of conventional construction materials and be water cooled as all physical and chemical processes mainly proceed far from the walls. The PDG is effectively cooled down from interior during the fill process, and can also be water cooled from the exterior with further use of the heated water for producing the low-temperature steam required for the operation process.”

  1. Title: “liquid hydrocarbon waste”, Section 2: “Organic waste”. Figure 1, it is hard to tell what is the waste exactly. Please indicate what is the waste…I believe the composition of the waste have a significant impact on the gasification outcome. For instance, if the waste is completely organic, can it still produce residues after gasification?

To address this comment, we have added the following sentence at the beginning of Introduction:

“A variety of carbon containing waste materials with different properties consisting predominantly of C, H, and O elements is commonly referred to as organic waste.”

  1. Section 3 and 4, please add more analysis of the results. For example, discuss the impact of the results rather than just display the data. In fact, Results and Discussion can be combined. Please reduce the description of the data. Instead, provide some discussion based critical thoughts. For example, how those trend from different models will impact future applications, etc.

To address this comment, we have combined the Results and Discussion sections and added some more clarifying sentences to the text.

  1. Equation (1). Is it Ok to use “C16H34” to represent any organic waste? A chemical with such a composition can be used as a fuel. Also, in life, it is difficult to find organic waste that does not contain any oxygen.

As mentioned in Section 2.2, n-hexadecane is often used as a physical and chemical surrogate of lubrication oil. Therefore, we use it as a waste oil (WO). Surely, WO can contain oxygen, water, chlorine, etc. This study is focused on the basics of the gasification process. Further studies will include the effect of other elements. To address this comment, we have added the following sentence to the Conclusions:

“Future work will be focused on the experimental implementation of the new technology of organic waste gasification with self-feeding of the pulsed detonation gun with the produced syngas. Further efforts will be also taken to include the various other properties of organic wastes into the thermodynamic analysis, like water, oxygen, and chlorine content.”

  1. Conclusion. Please combine the bullets.

Done

Round 2

Reviewer 2 Report

Comments and Suggestions for Authors

The manuscript is clearly improved and may be accepted for publication.

Reviewer 3 Report

Comments and Suggestions for Authors

I think the authors did a fair job revising the paper, now it can be published.